# Evaluating Different Storage Media for Identification of *Taenia saginata* Proglottids Using MALDI-TOF Mass Spectrometry

**DOI:** 10.3390/microorganisms9102006

**Published:** 2021-09-22

**Authors:** Tabea P. Wendel, Maureen Feucherolles, Jacqueline Rehner, Sven Poppert, Jürg Utzinger, Sören L. Becker, Issa Sy

**Affiliations:** 1Institute of Medical Microbiology and Hygiene, Saarland University, 66421 Homburg, Germany; tabeawendel@gmx.de (T.P.W.); jacqueline.rehner@uks.eu (J.R.); issa.sy@uks.eu (I.S.); 2Environmental Research and Innovation Department, Luxembourg Institute of Science and Technology, L-4422 Belvaux, Luxembourg; maureen.feucherolles@list.lu; 3Swiss Tropical and Public Health Institute, CH-4002 Basel, Switzerland; sven@poppert.eu (S.P.); juerg.utzinger@swisstph.ch (J.U.); 4University of Basel, CH-4003 Basel, Switzerland

**Keywords:** cestodes, diagnosis, helminth infections, matrix-assisted laser desorption/ionization time-of-flight (MALDI-TOF) mass spectrometry, neglected tropical diseases, taeniasis

## Abstract

*Taenia saginata* is a helminth that can cause taeniasis in humans and cysticercosis in cattle. A species-specific diagnosis and differentiation from related species (e.g., *Taenia solium*) is crucial for individual patient management and disease control programs. Diagnostic stool microscopy is limited by low sensitivity and does not allow discrimination between *T. saginata* and *T. solium*. Molecular diagnostic approaches are not routinely available outside research laboratories. Recently, matrix-assisted laser desorption/ionization time-of-flight (MALDI-TOF) mass spectrometry (MS) was proposed as a potentially suitable technique for species-specific helminth diagnosis. However, standardized protocols and commercial databases for parasite identification are currently unavailable, and pre-analytical factors have not yet been assessed. The purpose of this study was to employ MALDI-TOF MS for the identification of *T. saginata* proglottids obtained from a human patient, and to assess the effects of different sample storage media on the technique’s diagnostic accuracy. We generated *T. saginata*-specific main spectral profiles and added them to an in-house database for MALDI-TOF MS-based diagnosis of different helminths. Based on protein spectra, *T. saginata* proglottids could be successfully differentiated from other helminths, as well as bacteria and fungi. Additionally, we analyzed *T. saginata* proglottids stored in (i) LC–MS grade water; (ii) 0.45% sodium chloride; (iii) 70% ethanol; and (iv) 37% formalin after 2, 4, 6, 8, 12, and 24 weeks of storage. MALDI-TOF MS correctly identified 97.2–99.7% of samples stored in water, sodium chloride, and ethanol, with log-score values ≥2.5, thus indicating reliable species identification. In contrast, no protein spectra were obtained for samples stored in formalin. We conclude that MALDI-TOF-MS can be successfully employed for the identification of *T. saginata*, and that water, sodium chloride, and ethanol are equally effective storage solutions for prolonged periods of at least 24 weeks.

## 1. Introduction

The beef tapeworm, *Taenia saginata*, is a zoonotic cestode that can cause taeniasis, an intestinal infection in humans, and cysticercosis in bovines [1]. It is the most common and most widely distributed *Taenia* species. While humans are the definitive host, cattle serve as intermediate hosts for *T. saginata*. *Taenia solium* and *Taenia asiatica* are less frequently occurring species, with *T. solium* being of particular clinical relevance, as it gives rise to intestinal disease and the potentially fatal human (neuro-)cysticercosis [2]. Humans acquire intestinal *Taenia* infection through the consumption of raw or undercooked meat of infected animals. Intestinal taeniasis mainly causes mild and unspecific symptoms, such as weight loss and general malaise. More pronounced symptoms (e.g., diarrhea, abdominal pain, and nausea) are less frequent [3]. Severe complications, such as appendicitis or gall bladder perforation, have rarely been reported [4].

After the ingestion of infected bovine muscle tissue, a *Taenia* cysticercus develops within the human host’s intestine into an adult worm during a prepatency period of approximately 2 months, and produces eggs and gravid proglottids, which are shed with the feces. In settings with poor sanitation, eggs can spread through water, wind, or simply attach to vegetation. Cattle become infected by ingesting contaminated plants [5].

Taeniasis is considered a neglected tropical disease (NTD) [6]. In recent years, several studies carried out by the European CystiNet network and others investigated the global occurrence of taeniasis. It was found that *Taenia* tapeworms occur worldwide, and that *T. saginata* is particularly frequent in East, Southeast, and South Asia [7]. In Europe, taeniasis cases are reported in 12 out of 18 surveyed countries, with an estimated prevalence ranging from 0.02 to 0.67% [1]. As taeniasis is associated with poor sanitation, low-income settings, and understaffed meat inspectorates, the disease is also frequently reported from parts of the Middle East, Africa [8], and Central and South America [9]. However, prevalence estimates lack accuracy, as taeniasis is a non-notifiable disease in most countries, and as for many NTDs, public health campaigns pay little attention to this disease [10]. In 2007, it was estimated that at least 60 million people were infected with *T. saginata* [11]. However, the global burden of taeniasis, as expressed in disability-adjusted life years (DALYs), has yet to be determined [12].

The diagnosis of human taeniasis mainly relies on the direct visualization of proglottids, or the microscopic detection of eggs in stool samples [4]. In research settings, other methods are also used, such as stool-based polymerase chain reaction (PCR) assays or copro-antigen enzyme-linked immunosorbent assay (ELISA) tests, which detect specific secretory antigens in fecal samples [3]. However, these techniques have several limitations. While the commonly employed microscopy can be rapidly performed and does not require well-equipped laboratories, its sensitivity is low [4], and a species differentiation between *T. saginata* and *T. solium* is only possible if proglottids are shed in the feces, because the eggs of both species are indistinguishable [13]. The copro-antigen ELISA is characterized by a relatively low specificity, as studies carried out on samples stemming from cattle reported relatively high rates of cross-reactivity with related species of veterinary importance, such as *Taenia hydatigena* and *Taenia multiceps* [14]. PCR-based assays allow highly sensitive species identification, but are costly, rarely available outside research laboratories, and require specific technical expertise. Hence, there is a need for simple-to-use, accurate diagnostic methods for taeniasis, as the correct identification of *Taenia* infections at the species level is an important requirement for clinical management and contact screening, particularly in case of *T. solium* infections that pose the risk of human neurocysticercosis [15].

Matrix-assisted laser desorption/ionization time-of-flight (MALDI-TOF) mass spectrometry (MS) is an extensively validated diagnostic technique, which is nowadays routinely used in clinical microbiology laboratories for the species-specific diagnosis of bacteria and fungi in high-income countries [16]. Recently, several studies also reported MALDI-TOF MS, which analyzes pathogen-specific protein spectra to reach a specific diagnosis, as a suitable method for the identification of parasites [13], including helminths (e.g., *Fasciola* spp. [17], *Trichinella* spp. [18], and *Anisakis* spp. [19]). Besides high accuracy, the low cost of reagents needed for MALDI-TOF analysis in comparison to reagents required for PCR assays is a competitive advantage. However, there is uncertainty regarding the standardization of MALDI-TOF analytical protocols, and the effects of pre-analytical factors need to be elucidated. In this study, we utilized *T. saginata* proglottids to systematically assess whether the use of different sample storage media or the duration of storage affect the composition of the resulting protein spectra, and hence, the ability of MALDI-TOF MS to reach species-specific identification.

## 2. Materials and Methods

### 2.1. Ethics Statement

The *T. saginata* sample used in this study was obtained from an infected patient who sought routine diagnostic work-up for suspected parasite infection at the Swiss Tropical and Public Health Institute (Swiss TPH) in Basel, Switzerland. All procedures adhered to local laws and regulations.

### 2.2. Sample Collection

*T. saginata* proglottids were collected by an experienced medical laboratory technician from the stool sample of an infected patient at Swiss TPH in Basel. The specimen was stored in a freezer at −20 °C in 0.45% (*v*/*v*) sodium chloride solution. In October 2018, the sample was transferred to the Institute of Medical Microbiology and Hygiene in Homburg, Germany for further examination.

### 2.3. Study Design and Experimental Set-Up

Upon receipt at the Institute of Medical Microbiology and Hygiene in Homburg, the *Taenia* specimen was subjected to nucleic acid extraction, PCR, and partial sequencing for species-specific identification as *T. saginata*. Next, MALDI-TOF MS was carried out to generate protein spectral profiles, which were then transferred into an in-house database for MS-based identification of helminths. Subsequently, proglottids were put into different storage media and re-analyzed by MALDI-TOF MS after 2, 4, 6, 8, 12, and 24 weeks. At each time, the obtained spectra were compared to the initially measured spectra.

### 2.4. Molecular Diagnosis Using PCR and Partial Sequencing

For confirmatory molecular species identification, one proglottid of the *Taenia* specimen was thawed and subjected to DNA extraction using the DNeasy Blood and Tissue Kit (Qiagen GmbH; Hilden, Germany). In brief, a sample measuring approximately 1 cm was pounded into small pieces. Next, 180 µL of ATL buffer was added, the sample was vortexed, and 20 µL of proteinase K was added. The mix was vortexed and incubated at 56 °C in a thermomixer (Eppendorf; Hamburg, Germany) for 1 h. After incubation, the mix was vortexed again, and both 200 µL of AL buffer and 200 µL of 100% (*v*/*v*) ethanol were added. Subsequently, the DNeasy Mini column system (Qiagen; Hilden, Germany) was used for nucleic acid extraction, adhering to the manufacturer’s protocol.

For gene amplification, the partial mitochondrial cytochrome oxidase 1 gene (COX-1) was used to perform a PCR as previously described [20]. Specific forward (5′-CATCATATGTTTACGGTTGG-3′) and reverse (5′-GACCCTAATGACATAACATAAT-3′) primers were used to amplify a gene of around 350 base pairs (bp), utilizing a peqSTAR thermocycler (VWR; Radnor, PA, USA). In brief, the assay consists of 12.5 µL Hotstart Mix (Qiagen; Hilden, Germany), 0.5 µL of forward primer, 0.5 µL of reverse primer, 9.5 µL of water, and 2 µL of *Taenia* DNA. The cycling conditions comprised an initial denaturation step at 95 °C for 5 min, followed by 56 °C for 1 min, and 72 °C for 2 min. Then, 45 amplification cycles were performed, each consisting of a denaturation step at 95 °C for 30 s, annealing at 56 °C for 30 s, and elongation at 72 °C for 30 s. Afterwards, a final elongation step at 72 °C for 4 min was performed.

For sequencing of the generated amplicons, the Capillary Electophoretic GenomeLab genetic analysis system (Beckman Coulter; Brea, CA, USA) was used. Consensus sequences were created by editing and merging raw forward and reverse sequences, using the BioEdit© software version 7.2.5 (Tom Hall; Carlsbad, CA, USA). The consensus sequence was aligned with sequences deposited in the National Center for Biotechnology Information (NCBI) GenBank database for final identification.

### 2.5. Differential Sample Storage Conditions

*Taenia* proglottids were removed from the original storage medium (sodium chloride 0.45% (*v*/*v*)) and placed on a Petri dish. Using a sterile scalpel, individual proglottids were cut into small pieces of approximately 1 cm. Next, each specimen was placed into a 1.5 mL Eppendorf tube, and 1 mL of one of the following four different storage solutions was added: (i) sodium chloride 0.45% (*v*/*v*) (Merck KG; Darmstadt, Germany); (ii) ethanol 70% (*v*/*v*) (Merck KG); (iii) liquid chromatography (LC) MS grade water (Merck KG); and (iv) formalin 37% (*v*/*v*) (Merck KG). All samples were then stored at −20 °C in these media, before being consecutively subjected to MALDI-TOF MS after the aforementioned exposure periods. The experiment was carried out with 6 specimens for each storage medium, i.e., 24 proglottids in total.

### 2.6. MALDI-TOF Analysis

#### 2.6.1. Protein Extraction

Prior to analysis, each proglottid sample was thawed and cut into three equal parts. Each part was then transferred to a new tube for subsequent MALDI-TOF MS measurements. For protein extraction, we employed a previously developed protocol [17].

#### 2.6.2. MALDI-TOF Target Plate Preparation and Measurements

Using the protein extract, 1 µL of the supernatant was spotted onto the MALDI target plate. For each sample, eight specific spots on the target plate were used, as recommended by the manufacturer (MSP creation protocol V1.1; Bruker Daltonics; Bremen, Germany). After drying, 1 µL of α-cyano-4-hydroxycinamic acid (CHCA) matrix solution (Bruker Daltonics), composed of saturated CHCA, 50% (*v*/*v*) of acetonitrile, 2.5% (*v*/*v*) of trifluoracetic, and 47.5% (*v*/*v*) of LC–MS grade water, was added to each spot. A commercially available Bacterial Test Standard (BTS; i.e., *Escherichia coli* extract connected with two high molecular weight proteins) was used to calibrate the mass spectrometer. After drying at room temperature, the MALDI target plate was placed into the Microflex LT Mass Spectrometer (Bruker Daltonics; Bremen, Germany) for MALDI-TOF MS analysis. Each sample spot was measured four times to generate a total of 32 raw spectra (8 spots × 4). This procedure was carried out on two replicates on the same day (repeatability analysis), and on one additional replicate on a subsequent day (reproducibility analysis). Hence, a total of 96 raw spectra were acquired for each sample.

#### 2.6.3. MALDI-TOF MS Parameters

All measurements were performed using the AutoXecute algorithm in the FlexControl^®^ software version 3.4. (Bruker Daltonics; Bremen, Germany). For each spot, 240 laser shots (40 laser shots each using six random positions) were used to generate protein spectral profiles in linear positive ion mode. The laser frequency was 60 Hz, and a high voltage of 20 kV and pulsed ion extraction of 180 ns were employed. The mass charge ratio range (m/z) was measured between 2 and 20 k Da.

#### 2.6.4. Spectral Analysis, MSP Creation, and Clustering Analysis

All raw spectra were analyzed with the FlexAnalysis^®^ software version 3.4 (Bruker Daltonics; Bremen, Germany). To improve the spectral quality, raw spectra were edited by removing all flatlines and outlier peaks. The intensities were smoothened, and baseline subtraction was performed, as appropriate. Peak shifts within spectra were also edited when they exceeded 500 ppm. Following these steps, replicates containing at least 22 remaining spectra were maintained, and the measurement was repeated if these conditions were not reached.

The edited spectra of the initial *Taenia* sample were used to create a species-specific main spectral profile (MSP), utilizing the automated function of the MALDI Biotyper Compass Explorer^®^ software version 4.1 (Bruker Daltonics; Bremen, Germany). The newly created *Taenia* MSP was added to a previously developed in-house database with several species, including cestodes (e.g., *Diphyllobothrium* spp.), nematodes (e.g., *Ascaris* spp.), and trematodes (e.g., *Fasciola* spp.), for helminth identification, and served as a reference spectrum for comparative analysis under different storage conditions. Subsequently, a clustering analysis was performed on the edited spectra obtained after 2, 12, and 24 weeks using the BioNumerics^®^ software version 7.6 (Applied Maths N.V.; Sint-Martens-Latem, Belgium). A dendrogram was generated using an unweighted pair group method with the arithmetic mean (UPGMA), and a curve-based similarity matrix was calculated using Pearson correlation. A principal components analysis (PCA) and a discriminant analysis were carried out using quantitative values.

#### 2.6.5. MALDI-TOF Identification Parameters

All measured spectra were initially analyzed using the official Bruker Taxonomy Database designed for bacteria and fungi, containing 8936 MSPs, which is routinely used in clinical microbiology laboratories, to detect possible contamination with bacterial or fungal organisms. Next, protein spectra were analyzed by a combination of this official Bruker database (Bruker Taxonomy) and the previously developed in-house helminth database with around 98 MSPs, including the MSP of the initially analyzed *Taenia* proglottid. The reliability of identification was evaluated by log score values (LSVs), which were generated by MALDI-TOF MS. We followed the LSV thresholds used in routine microbiology for the identification of bacteria and fungi, i.e., LSVs ≤1.69, indicating an unreliable identification; LSVs ranging between 1.70 and 1.99, indicating an accurate genus and probable species identification; and LSVs ≥2.0, suggesting a reliable species identification.

## 3. Results

### 3.1. Molecular Identification of Taenia Proglottids

PCR and sequencing of the initial *Taenia* proglottid sample using primers of the COX1-gene confirmed the species diagnosis. An analysis using NCBI GenBank showed 100% sequence homologies with a previously described *T. saginata* sequence (reference accession number: MT074048.1). The sequence of our *Taenia* sample was deposited in the GenBank database (accession number: MZ720823).

### 3.2. Comparative MALDI-TOF MS Analysis after Different Storage Periods

#### 3.2.1. Protein Spectra and LSV Analysis

A representative protein spectral profile for each storage medium is displayed in Figure 1. High peak intensities were observed and reached up to 1.0 × 10^4^ arbitrary units (a.u.). With regard to the position and the intensity of the measured peaks, *Taenia* samples stored in LC–MS grade water, ethanol, and sodium chloride showed a similar profile to the original sample, with no significant changes over time. For samples stored in formalin, no protein spectra were found at any time point.

For all samples, the commercially available MALDI-TOF database for the identification of bacteria and fungi did not yield a reliable identification, with an LSV of 1.37 for the bacterium *Arthrobacter monumenti* being the highest score. When submitting the spectra to a combination of the commercially available and in-house helminth databases, a correct identification was achieved in 97.2%, 99.7%, and 99.0%, for samples stored in sodium chloride, ethanol, and LC–MS grade water, respectively, with LSVs ranging between 2.53 and 2.57. No identification was achieved for spectra of *Taenia* proglottids stored in formalin (Table 1).

When analyzing identification patterns over time, a high LSV (≥2.3) was constantly observed at all measurements for each storage solution, except formalin. Small fluctuations of LSVs were found for all storage solutions, with slightly more fluctuation in the sodium chloride medium (Figure 2).

#### 3.2.2. Cluster Analysis

Cluster analysis to display the relatedness of the *Taenia* proglottids stored in sodium chloride, ethanol, and LC–MS grade water showed that all these proglottids clustered together and showed relatedness levels >85% (Figure 3). Subsequent statistical analyses (both PCA and discriminant analysis) performed on the summary spectra of *T. saginata* proglottids did not show specific differences pertaining to the different preservation media or the duration of storage (Figure 4), thus indicating an almost identical pattern of the protein spectra.

## 4. Discussion

The purpose of this study was to determine whether MALDI-TOF MS can be used as a diagnostic tool for the identification of *T. saginata* proglottids, and whether the use of different storage media may affect the technique’s diagnostic accuracy. We found that *T. saginata* can be diagnosed by MALDI-TOF MS, and that its protein spectral analysis allows for reliable differentiation from other helminths, bacteria, and fungi. Indeed, *T. saginata* was consistently identified correctly in ≥97% of cases if LC–MS grade water, ethanol, or 0.45% sodium chloride was used as a storage solution, with no changes over time for storage periods of up to 24 weeks. Notably, preservation in 37% formalin did not allow for subsequent MALDI-TOF MS examinations.

Our findings might have important implications for future helminth diagnosis in epidemiologic studies. Indeed, MALDI-TOF MS is a widely used diagnostic tool in microbiologic routine diagnosis [21,22], which will also be increasingly available in laboratories of low- and middle-income countries [23]. Besides the identification of bacteria and fungi, this technique has also been successfully used for the differentiation of ticks and fleas [24], mosquitos [25], lice [26], and more recently, different helminths of medical and veterinary importance [13]. Hence, MALDI-TOF MS could also be employed for confirmatory testing of helminths in reference laboratories, for example, when no unambiguous identification is reached by conventional methods. However, prolonged transport periods of samples from peripheral healthcare centers to such reference laboratories are likely to be expected, and hence, information on the most appropriate sample storage media is key to ensure a reliable analysis by MALDI-TOF MS. In this context, it is important to note that different protocols were utilized in studies conducted thus far, as there is no consensus on the most suitable storage media. For the identification of *Fasciola* spp. [17], cyathostomins [27], and lice [26], 70% (*v*/*v*) ethanol was used as a storage solution, while studies on *Anisakis* spp. [19], *Dirofilaria* spp., and *Ascaris* spp. [28] employed a sodium chloride solution, which was sometimes even supplemented with antibiotics to prevent bacterial contamination. Nebbak et al. [24] analyzed the effects of different storage conditions on the identification of arthropods. The authors concluded that the immediate freezing of samples without the addition of any fixative might be the best approach, closely followed by storage in 70% (*v*/*v*) ethanol at room temperature.

Only a few investigations have assessed the potential effects of different storage conditions on the subsequent MALDI-TOF MS-based identification of helminths. A study focusing on *Trichinella* spp. did not observe significant differences in identification rates when either freezing without any fixative or using 70% ethanol. Indeed, only minor alterations of measured peak intensities were reported, but no change in peak patterns or obtained LSVs [29]. In our study, LC–MS grade water, ethanol, and sodium chloride were equally effective in maintaining a high quality of protein spectra for up to 24 weeks, with correct identification rates ranging from 97.2% for sodium chloride to 99.7% for ethanol at −20 °C. In addition, a statistical analysis of the protein spectra did not reveal fixative-related clusters, thus confirming that all three media can be equally used as storage solutions for *T. saginata* proglottids until MALDI-TOF MS is carried out. Notably, preservation in formalin and subsequent protein extraction using formic acid and acetonitrile impeded any MALDI-TOF-based identification, and hence, should not be employed. This observation is not surprising, as formalin induces considerable molecular cross-linking that may change protein structures [30].

Several limitations restrict the generalizability of our findings. First, the proglottids used in this study were originally stored in sodium chloride for 12 months, before being assigned to the different storage media. Hence, future studies should employ fresh specimens. However, data from a study on suitable buffers for MALDI-based screening of biochemical targets suggest no concerns with regard to the use of sodium chloride [31]. The results obtained in this study may confirm this fact. Second, we only assessed potential effects on *T. saginata*; the in-house database is restricted as it does not contain other *Taenia* species, such as *T. solium*. While it is unlikely that other helminth species would react differently, a broader validation on similar cestodes—most importantly *T. solium*—as well as on nematodes and trematodes is desirable. Specifically, all developmental stages of helminths, including their eggs, should be subjected to MALDI-TOF-based examinations. Third, we compared the effects of different media stored at −20 °C, while future research should also assess the potential effects of storage at different temperatures.

## 5. Conclusions

We conclude that MALDI-TOF MS is a promising tool for the rapid and accurate identification of *T. saginata* proglottids. Samples can be reliably identified after prolonged storage in LC–MS grade water, sodium chloride solution, and ethanol, while formalin cannot be used as a fixative for later MALDI-TOF MS analysis.

## Figures and Tables

**Figure 1 microorganisms-09-02006-f001:**
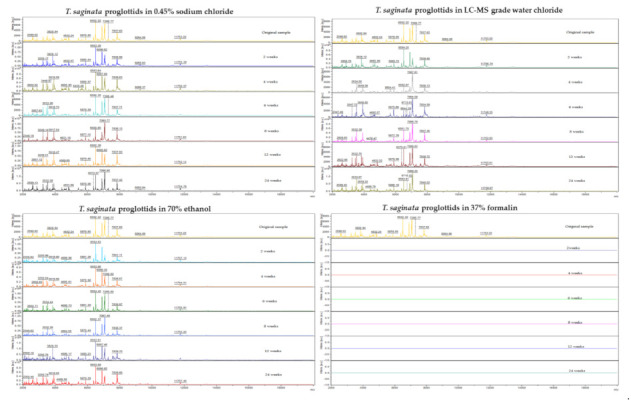
Protein spectral profiles of *Taenia saginata* proglottids. The peaks obtained when measuring the original sample, and protein profiles after prolonged storage in four different media, are displayed. *X*-axis, mass-to-charge ratio of (*m*/*z*); *Y*-axis, peak intensities of ionized molecules; a.u., arbitrary unit.

**Figure 2 microorganisms-09-02006-f002:**
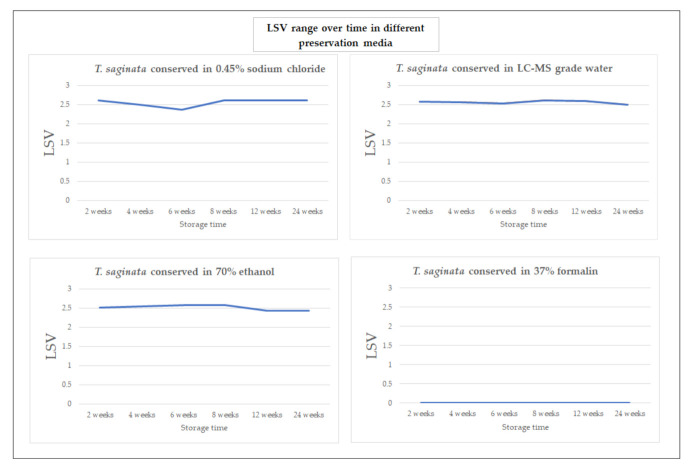
Average LSVs of protein spectra stemming from *Taenia* proglottids in different preservation media during a 24-week observation period. Spectra were identified using a combination of Bruker Taxonomy and an in-house helminth database.

**Figure 3 microorganisms-09-02006-f003:**
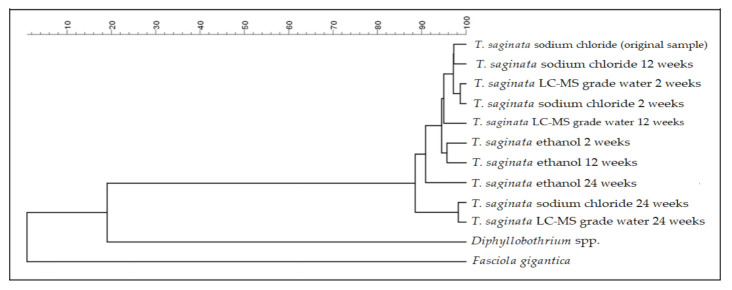
Dendrogram derived from a clustering analysis to assess and compare the different protein spectra of *Taenia saginata* proglottids stored in three storage media for different time periods. The cestode *Diphyllobothrium* spp. and the trematode *Fasciola gigantica* were added as outgroup samples.

**Figure 4 microorganisms-09-02006-f004:**
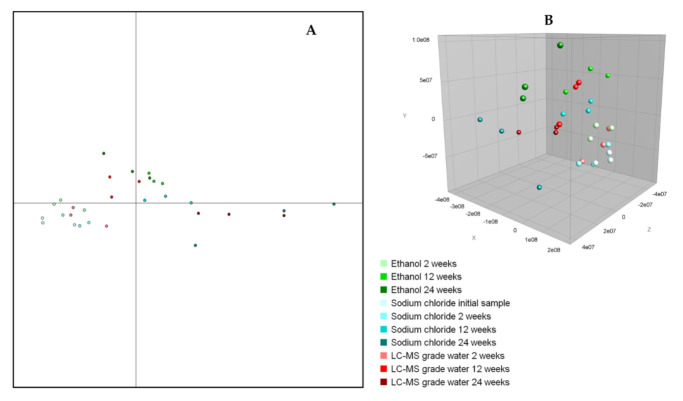
Discriminant analysis and principal components analysis (PCA) of *Taenia saginata* proglottids stored in different preservation media for different time periods. Each storage medium is depicted with a different color. Both statistical analyses indicate that the clusters are highly related and cannot be separated. (**A**) Two-dimensional view of the discriminant analysis. (**B**) Three-dimensional view of the PCA.

**Table 1 microorganisms-09-02006-t001:** Identification of *Taenia saginata* proglottids stored in different storage media (**A**) using Bruker Taxonomy, the commercially available database for bacteria and fungi, and (**B**) using a combination of Bruker Taxonomy and an in-house helminth database.

(A)					
Sample Preservation Medium	Number of Samples	Number of Spectra	Bruker Taxonomy Database		
			Correct Identification	Average LSV	Most Frequently Suggested Result
0.45% sodium chloride	6	560	0%	1.38	*Arthrobacter monumenti*
70% ethanol	6	574	0%	1.39	*Arthrobacter monumenti*
LC–MS grade water	6	570	0%	1.38	*Arthrobacter monumenti*
37% formalin	6	0	0%	0	None
(**B**)					
**Sample Preservation Medium**		**Number of Spectra**	**Combination of Bruker Taxonomy and In-House Helminth Database**		
	**Number of Samples**		**Correct** **Identification**	**Average LSV**	**Most Frequently Suggested** **Result**
0.45% sodium chloride	6	560	97.2% (560/576)	2.54	*T. saginata* proglottid
70% ethanol	6	574	99.7% (574/576)	2.53	*T. saginata* proglottid
LC–MS grade water	6	570	99.0% (570/576)	2.57	*T. saginata* proglottid
37% formalin	6	0	0%	0	-

## Data Availability

The data presented in this study are available on reasonable request from the corresponding author.

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
