# Peer review of "Evaluating Different Storage Media for Identification of Taenia saginata Proglottids Using MALDI-TOF Mass Spectrometry"

_microorganisms, 2021, doi:10.3390/microorganisms9102006_

Round 1

Reviewer 1 Report

22 - consider the change to ..for the identification

42 - “intermediate host”  seems to be missing a determiner before it

83-   please consider the change to: for taeniasis, as the correct identification of

88 - please consider the change to: high-income countries

206 - please consider the change to: detect possible contamination

259  - please consider the change to: Cluster analysis to display...

263-  sentence contain double negative inclusion: did not and  neither

285 - please consider the change to: with no changes overtime for

290 - please consider the change to: Besides the identification of

292 - please consider the change to: technique has also been successfully used for

299 - please consider the change to: a reliable analysis by MALDI-TOF

302 - please consider change to: ethanol was used as a storage solution

307 - please consider the change to: might be the best approach,

309 - please consider the change to: Only a few investigations assessed the potential effects

311 - please consider the change to: differences in identification

A very interesting article solving MALDI TOF basic problems.  The effect of formalin on proteins is well known. Extraction from formalin-preserved samples is currently the focus of numerous scientific works. Therefore, an unequivocal statement that the identification of formalin-preserved samples is impossible should refer only to the extraction used. I suppose the authors to consider this issue.

Author Response

Please see our detailed point-by-point response, which is attached to the revision of our manuscript.

Reviewer 2 Report

In the submitted paper authors evaluating different storage media for identification of Taenia saginata proglottids using Maldi-TOF Mass Spectrometry. Moreover, they investigate the potenetail application of the MALDI-TOF Mass Spectrometry to the detection of Taenia saginata proglottides which, if are present in the stool, are quite easily detectable in it and moreover permit for distinguish T. saginata from T. solium by the using of microscope methods. In turn, eggs of this helminths, that’s different. Despite, this manuscript contains some new interesting results and in my opinion may be considered to publish in Microorganism, but after major revision see below: 

General comments

The starting material were proglottids stored for a period time in a freezer at -20o C in 0.45% (v / v) sodium chloride solution. So, the control material was preserved in this media for a period time (authors write it was several months) before performing the presented studies. Why then authors are showing on the Figure 1 the spectra profiles of Taenia saginata proglottids profiles after prolonged storage in this media after 2-24 weeks if they had been in it for much longer time? The authors are aware of this limitation in they results but unfortunately it is not enough. In my opinion in this study the control sample should be a frozen proglottids without addition of any fixative. Then only we can study the influence of different media and the temperature for the protein spectra, because temperature itself may also have some influence for them. Moreover, there is a lack of period of time in a original sample. In addition the presented protein spectral profiles of the original sample should be presented the same as the remaining, that the reader could compare it together.  I suggest also to consider the validity of the presentation of Figure 3 in the tekst of the manuscript. The authors also write that this method is a promising for a rapid and accurate identification of T. saginata proglottids. I think that the note about the cost of such analysis comparing to the PCR analysis would be valuable.

Detailed comments

  1. Line 120, there is „DNeasy Bood and Tissue Kit”. It sholud be „ DNeasy Blood and Tissue Kit”
  2. Line 241, there is „T.saginata”. It should be „Taenia saginata” and this species name shold be written in italics.
  3. Line 255, „Taenia” should be written in italics.
  4. Lines 269-271, the species names should be written in italics.
  5. Line 274, the species name should be written in italics.

Author Response

(The authors gave the same response as above.)

Round 2

Reviewer 2 Report

I my opinion this revised version of the manuscript may be publish in „Microorganism” after few minor revision see below:

1.    I would suggest to add in the disscusion section the one short sentence „The results obtained in this study may confirm this fact”, after the senetence „However, data from a study on suitable buffers for MALDI-based screening of biochemical targets suggest no concerns with regard to the use of sodium chloride [31]”.

2.    In the Figure 2, I would suggest to change the order of the charts similar to the Figure 1.

3.    Lines 244 and 277 „ Figure 1”, Figure 4” should not be written in italics.

Author Response

Please see our point-by-point response in the attached document.
